# Implementing a New Electronic Health Record System in a University Hospital: The Effect on Reported Medication Errors

**DOI:** 10.3390/healthcare10061020

**Published:** 2022-05-31

**Authors:** Carita Lindén-Lahti, Sanna-Maria Kivivuori, Lasse Lehtonen, Lotta Schepel

**Affiliations:** 1Quality and Patient Safety Unit, HUS Joint Resources, Helsinki University Hospital and University of Helsinki, 00029 Helsinki, Finland; lotta.schepel@hus.fi; 2HUS Pharmacy, Helsinki University Hospital and University of Helsinki, 00029 Helsinki, Finland; 3Division of Pharmacology and Pharmacotherapy, Faculty of Pharmacy, University of Helsinki, 00014 Helsinki, Finland; 4Helsinki University Hospital and University of Helsinki, 00029 Helsinki, Finland; sanna-maria.kivivuori@hus.fi; 5HUS Diagnostic Center, Helsinki University Hospital and University of Helsinki, 00029 Helsinki, Finland; lasse.lehtonen@hus.fi

**Keywords:** electronic health record system, electronic medication management system, EPIC, APOTTI, medication error reporting, prescribing errors, medication safety, patient safety

## Abstract

Closed-loop electronic medication management systems (EMMS) have been seen as a potential technology to prevent medication errors (MEs), although the research on them is still limited. The aim of this paper was to describe the changes in reported MEs in Helsinki University Hospital (HUS) during and after implementing an EPIC-based electronic health record system (APOTTI), with the first features of a closed-loop EMMS. MEs reported from January 2018 to May 2021 were analysed to identify changes in ME trends with quantitative analysis. Severe MEs were also analysed via qualitative content analysis. A total of 30% (n = 23,492/79,272) of all reported patient safety incidents were MEs. Implementation phases momentarily increased the ME reporting, which soon decreased back to the earlier level. Administration and dispensing errors decreased, but medication reconciliation, ordering, and prescribing errors increased. The ranking of the TOP 10 medications related to MEs remained relatively stable. There were 92 severe MEs related to APOTTI (43% of all severe MEs). The majority of these (55%, n = 53) were related to use and user skills, 24% (n = 23) were technical failures and flaws, and 21% (n = 21) were related to both. Using EMMS required major changes in the medication process and new technical systems and technology. Our medication-use process is approaching a closed-loop system, which seems to provide safer dispensing and administration of medications. However, medication reconciliation, ordering, and prescribing still need to be improved.

## 1. Introduction

Medication safety is one of the key areas of patient safety [1,2]. A medication error (ME) is any preventable event that may cause or lead to inappropriate medication use or patient harm [3]. As recognised globally [2], MEs are one of the most prevalent types of errors reported to the error reporting system in Finland [4]. Prescribing, administering, monitoring, and transition of care are the most error-prone phases for severe MEs in the medication process [2,5,6]. Furthermore, high-alert medications, which bear a heightened risk of causing significant patient harm when used in error, should attract special focus [2,7,8].

Closed-loop Electronic Medication Management Systems (EMMS) have been seen as a potential technology to prevent medication errors and enhance the quality of the medication process, although the research on them is still limited [9,10]. Electronic medication management is a broad term covering all computer systems involved: it refers to a closed-loop system that encompasses prescribing, administration, pharmacy verification, smart infusion pumps, automated dispensing cabinets, barcode medication administration, and anything that has electronic medicine datasets or encompasses medication management processes [11,12]. Electronic health record (EHR) systems should enable the technology to achieve a closed-loop medication management process with EMMS. This goal was one of the aims when Helsinki University Hospital (HUS) decided to change its EHR system to an EPIC-based APOTTI system.

Some previous studies have evaluated EPIC’s impact on patient and medication safety, but they have mostly been from the perspective of an individual department or patient group. Other hospitals implementing EPIC have reported some new challenges in patient safety [13,14]. Findings of safer processes and improved documentation have been published [15,16]. There is a need to understand how EHRs based on closed-loop medication management systems affect medication errors and which errors may be new medication safety concerns. This study aimed to describe the changes in reported medication errors during and after implementing an EPIC-based EHR system (APOTTI), with the first features of a closed-loop EMMS in HUS.

## 2. Materials and Methods

### 2.1. Study Setting

This study was conducted at Helsinki University Hospital (HUS), which provides secondary and tertiary care via 23 hospitals for a population of 1.6 million in the capital area of Finland. HUS implemented an EPIC-based EHR system (APOTTI) in four phases (Go-Lives, GLs): GL1 in November 2018; GL2.1 in February 2020, GL2.2 in October 2020, and GL3 in April 2021, but the latter phase was not related to medication management. APOTTI enables a closed-loop electronic medication management system (EMMS). The key changes in the HUS medication management process before and after using APOTTI are described in Table 1.

A voluntary electronic reporting system for patient and medication safety incidents (HaiPro) has been used since 2007. It is currently in use in more than 60% of all hospitals and other healthcare units in Finland [17]. In HUS, HaiPro was introduced in 2007 and extended to all departments in 2011. In the HaiPro report form, the reporter is requested to specify the nature of an incident (reached patient, near miss, other proactive observations), comment with an open-text field on the circumstances and contributing factors to an error, and share ideas on how the error could be prevented in the future. These features make HaiPro comprehensive and system-oriented. From the HaiPro database, it is possible to search which medications and Anatomical Therapeutic Chemical (ATC) Classification groups (5th level, chemical substances) [18] are related to MEs in cases where a specific medication has been reported.

The incident reports are based on narratives coded according to the stages of the medication use process in the units by staff members, usually nurses responsible for managing the ward, trained to do the coding. They also determine the consequences for patients and the risk category (I–V) of an error. Errors that caused or could have caused (also near misses and other proactive observations) severe patient harm or were categorised in the highest risk classes (IV–V) go automatically to the quality manager of the department, who reviews the coding and accepts severe reports for a root cause analysis process. The definitions of severe patient safety incidents used in previous studies vary greatly [19]. The definition used by HUS for severe MEs refers to errors that caused severe harm or have the potential to cause severe harm.

Some changes were made to ME subtypes in the HaiPro tool during the study time. The documenting error subtype was found to be impractical after implementing the APOTTI system, when structured documentation became part of each stage in the medication-use process. Hence, documenting of the error subtype was removed and embedded in other ME subtypes (e.g., ordering and prescribing, administration and dispensing errors) in the beginning of 2021. New subtype medication reconciliation at admission was added at the same time (earlier this was included in documenting errors). The monitoring error subtype was added in the beginning of 2021.

### 2.2. Data Collection and Analysis

ME data from January 2018 to May 2021 (six months after GL 2.2) were imported to Microsoft Excel from the HaiPro system. This period was chosen because we needed the data rapidly for continuing development of APOTTI. The coded nature of MEs, ME subtypes, and the most commonly reported (TOP) active substances and ATC groups related to MEs were analysed. The Institution for Safe Medication Practices’ (ISMP) high-alert medications for acute and ambulatory care settings were identified [7]. The ISMP’s lists were chosen because they are widely used high-alert medication lists internationally. Drug consumption data were derived from the hospital pharmacy register to check whether the changes in drug-specific MEs were related to changes in drug consumption. If there were clear changes in TOP20 medications related to MEs, the narratives of those ME reports were analysed with qualitative content analysis [21] to find out possible reasons for changes by one researcher (L.S.).

Severe MEs were manually searched from all severe patient safety incidents (caused severe patient harm and/or risk category IV–V) reported in the HUS HaiPro system. Reports which included MEs and were related to the APOTTI system were included. Severe MEs were analysed separately with qualitative inductive content analysis [21] into three new categories: (1) technical failures and flaws in APOTTI; (2) proper use or user’s knowledge and skills; and (3) L.S. conducted both of these analyses, and if there was the possibility of the interpretability of the category, it was categorised as a consensus of two researchers (L.S. and C.L-L). The medications involved in severe MEs were also identified.

### 2.3. Study Ethics

This study received a research permit from the HUS Joint Authority (HUS/157/2020). Ethical approval was not needed as the study data did not include any patient data.

## 3. Results

During the study period (January 2018–May 2021), 30% (n = 23,492) of all reported patient safety incidents (n = 79,272) were related to MEs in HUS: in 2018 (30%; n = 6857/22,577), 2019 (30%; n = 7261/24,179), 2020 (29%; n = 6548/22,453) and 1–5/2021 (28%, n = 2826/10,063). Patient incident reporting activity in HUS increased annually during the past few years, but in 2020 it decreased for the first time. MEs were the most reported error type in all four years considered, and the reporters of MEs during the study period were 77% nurses, 6% allied health team members (e.g., pharmacists), and 5% doctors. The nature of the MEs remained relatively stable: 48–63% of the reported errors reached patients, and the rest, 37–52%, were near misses and other proactive observations (Figure 1).

Different implementation phases (APOTTI GLs) momentarily increased the number of reported MEs, which soon decreased to the same level as before (Figure 1). During the study period, the number of reported administration and dispensing errors decreased, and ordering and prescribing errors increased (Figure 2). The main implementation phases (GL2.1. and GL2.2.) induced clear peaks in some reported contributing factors of MEs: working environment and resources, training introduction and skills, and communication and flow of information were highlighted in the GL2.1, whereas training, introducing and skills, and devices, equipment, and IT-systems were noted in the GL2.2 (Figure 3).

### 3.1. TOP Medications Related to MEs

TOP10 medications related to the reported MEs remained relatively stable during 2018–2021, but the TOP20 ranking revealed some changes (Figure 4). A notable change was seen in MEs related to enoxaparin versus other low-molecular-weight heparins (LMWHs), e.g., dalteparin and tinzaparin in 2021. However, this change is a consequence of changes in the hospital’s formulary. Enoxaparin has been the primary LMWH for several years, but at the end of 2020 an enoxaparin biosimilar was chosen for the formulary instead of the proprietary drug in competitive tendering. This change increased the use of MEs related to dalteparin. According to report narratives, MEs related to LMWHs are often related to ordering during and after surgical procedures, and problems are related to duplicate orders (verbal orders during surgery documented by operative nurses and written orders made by surgeons afterwards), or defining administration times as a part of the ordering process.

Another notable change to MEs related to levothyroxine, ranked 14th in 2018 vs. 5th in 2021 (Figure 4), while consumption decreased. According to report narratives, this increase is also related to the ordering process: the prescribed levothyroxine dose usually varies every other day, e.g., x mg on Mondays, Wednesdays, Fridays and Sundays and y mg on Tuesdays, Thursdays, and Saturdays. This kind of order was possible for one order in our earlier patient information system. However, in APOTTI, two different orders must be linked together (one order for Mondays, Wednesdays, Fridays and Sundays and another for Tuesdays, Thursdays and Saturdays). Narratives revealed that MEs related to this feature are common, especially when the reconciled home medication list is changed to hospital orders, because these orders require new orders linked together (home medications cannot just be continued to a hospital order).

Furthermore, this feature is not only related to ordering drugs with doses that vary every other day, but also drugs for which the dosage varies during the day. This situation was also highlighted in report narratives related to bisoprolol (e.g., dose 2.5 mg in the morning and 5 mg in the evening) in 2020 when its ranking rose (Figure 4), which was one reason why MEs related to levodopa and decarboxylase inhibitors appeared in the TOP20 ranking in 2021. At the same time, the consumption of bisoprolol, levodopa, and decarboxylase inhibitor remained stable. By contrast, MEs related to quetiapine decreased (Figure 4), while its consumption clearly decreased during the study time.

### 3.2. Severe MEs

After the first APOTTI Go-Live (10/2018-5/2021), 96 severe MEs were reported which related to medication use and use of the APOTTI system (2018: n = 3, 2019: n = 35, 2020: n = 44, 1–5/2021: n = 14). This was 43% (n = 96/225) of all severe MEs during that time. A total of 57% (n = 55) of the severe MEs related to APOTTI reached patients, 25% (n = 24) were near misses, and 18% were (n = 17) other proactive observations (not yet a near miss). Nurses made 45% (n = 45), physicians 38% (n = 36), and other staff (e.g., pharmacists) 18% (n = 17) of the severe reports. The reporting activity of physicians was higher with APOTTI related to all severe MEs (approximately 21% in the study period). Severe MEs were related to (1) use and user skills, 55% (n = 53); (2) technical failures and flaws, 24% (n = 23); and (3) both of these, 21% (n = 20). Subtypes of these varied considerably. Ordering and prescribing were common coded subtypes in severe MEs related to use, user skills, and training. Patient data management and documenting errors were highlighted in errors related to both technical failures and use.

Usually, severe MEs were linked to several medications (29%, n = 28), e.g., errors or problems in the entire medication list or medication reconciliation, including ordering and prescribing at admission, transitions, and discharge. Specific common medication groups linked to severe MEs were anticoagulants and antitrombothics (20%, n = 19), drugs affecting the nervous system (15%, n = 14), antibiotics (10%, n = 10), antineoplastic and immunomodulating agents (8%, n = 8), and cardiovascular drugs (5%, n = 5) or electrolyte infusions (3%, n = 3). The TOP medications were LMWHs (n = 5), cefuroxime (n = 5), vancomycin (n = 4), oxycodone (n = 4), abixaban (n = 3), warfarin (n = 3), amiodaron (n = 3), morphine (n = 2), methotrexate (n = 2), and tacrolimus (n = 2). The majority of these are regarded as high-alert medications.

## 4. Discussion

### 4.1. Changes in the ME Report Profile

As seen in this and previous studies, medication errors increase initially when a new EHR system is implemented [13]. This seems to be a normal phase in EHR system implementation, which organisations should be prepared for beforehand. During the study period, the global COVID pandemic seemed to reduce overall reporting activity, which was also seen in our data. Before implementing APOTTI, nurses mainly performed the medication error reports. As many problems with the new system were affiliated with ordering and prescribing, the reporting activity of the physicians increased. They were very active in reporting severe MEs related to APOTTI (38% vs. 5% of all MEs and approximately 21% of other severe MEs). Physicians’ higher reporting activity related to severe MEs is in line with earlier studies [22,23].

In this study, the most remarkable change after the new EHR system implementation can be seen in the profile of medication error types. This is probably a cause of the major changes in the medication process (Table 1), but also some of the medication safety problems became more visible with the new EHR system (e.g., medications not previously properly reconciled) [24]. Furthermore, some changes are due to the renovations of ME classification in our reporting tool. Previous studies have also described how EMMS implementation may cause some new safety risks, and the implementation of new technologies may cause socio-technical errors [25,26,27].

After the implementation of APOTTI, the TOP10 medications related to MEs remained relatively stable, which means that we still need to find other actions to enhance their safe use. The use of APOTTI decreased some error types for TOP10 medications, but also created new risks related to these medications. High-alert medications were common in the TOP10 medications, especially in severe cases, highlighting the importance of medication reconciliation and the need for new safety defences [7,8].

### 4.2. Creating a Structured Home Medication List Is Challenging

A key change with APOTTI use was the structured home-medication documentation, which was earlier written in free text at admission (Table 1). In APOTTI, the home-medication list is linked to the Finnish Kanta Prescription Centre, which holds electronic prescriptions [20]. A mandatory e-prescription system is helpful when home medications are reconciled, but Kanta still has some major deficiencies: (1) the data of the prescriptions are not entirely structured; (2) the Prescription Centre includes a lot of prescriptions that are not in use, and (3) it does not include data from over-the-counter (OTC) medications or dietary supplements that a patient is using. These make the medication reconciliation process at admission very laborious and prone to errors. The data must be checked with the patient to identify active e-prescriptions, list the OTC products, and then manually change all this information into a structured form, which is extra work compared to before APOTTI. Mandatory documenting of home medication lists has increased the number of clinical pharmacists working at the admission stage in HUS (e.g., emergency departments and ambulatory clinics), which has been shown to be beneficial [24,28,29].

### 4.3. Structured Ordering Process Still Needs Special Attention

The reconciled home medication list is used as a base for in-patient orders and the medication list at admission. Orders that need linking (e.g., levothyroxine and other examples described in the results section) cannot be continued. Physicians need to know and to remember to complete new linking in-patient orders. Linking orders is also challenging when new in-patient orders are done later during hospitalisation. Physicians would have needed more training on the structured ordering process before, during, and after implementation. This can be seen in the increased number of ordering and prescribing errors, and severe MEs related to this stage were commonly associated with use and user skills. The essential role of training has also been seen in previous studies, and the use of general training, instead of customised training based on local needs, can be a barrier to the adoption of EMMS [25,30,31].

Before implementing APOTTI, there were high hopes for a more advanced computerised decision support system (CDSS), especially regarding the structured ordering and prescribing stage. However, these results show that it is not yet sufficient to rely on alone. Our physicians felt massive alert fatigue, which can lead to overriding alerts [32,33]. Hence, we still need to optimise alerts and find a balance with them. There were also discussions that pharmacist-led medication reviews would no longer be needed after the implementation of APOTTI and its CDSS, which now seems incorrect. There is in fact a higher need for pharmacists to verify orders, even though this was not common in Finland earlier. APOTTI is the first EHR system in Finland to provide a system-assisted technical workflow for pharmacists’ order verification. HUS Pharmacy started to provide this service in its first units in 2020.

### 4.4. Implementing Barcode Scanning Is Beneficial

After HUS implemented barcode scanning in preparing, dispensing, and administering medicines with APOTTI, fewer ME reports were related to those activities. However, many units in HUS still do not use barcodes (e.g., operating, recovery and emergency rooms, and ambulatory units), and the scanning rates are not high enough. This needs to be addressed, but we see positive signs in medication safety. The utility of barcode-assisted medication administration (BCMA) has been recognised in previous studies [9,10,34].

Using APOTTI required not only introducing new BCMA technology and integrations but also major changes to the process, workflows, and protocols. Preparing and dispensing medicines in a timely manner (maximum 2 h before administration) and using only an electronic medication list before the actual GL was crucial to support the efficient implementation of BCMA. Workflows for closed-loop medication management and the BCMA process rely on correct orders, which may also explain why the reporting of ordering errors increased after implementation. Before APOTTI, nurses could administer medicine regardless of the original order. However, preparedness for the new dispensing, preparing, and administration process with bar-code scanning was successful in our hospital.

### 4.5. Other Lessons Learned in Implementing a New EHR System from a Medication Safety Perspective

As HUS consists of multiple hospitals, dividing an EHR system implementation into multiple phases was a successful approach. When APOTTI was implemented in the first hospital in 2018, it was known and agreed that the system had many unsolved problems which were supposed to be developed as it was used. This minimum viable product strategy [35] has many disadvantages, especially regarding patient and medication safety. Although EMMS typically require continuous development [27], testing and evaluating medication workflows and usability with a higher priority before implementation could have prevented many medication safety problems that we encountered [36,37]. However, between GLs, we had an opportunity to resolve the medication safety problems that were noticed and train our professionals in response. The new medication reconciliation process was a good example of this: we had not allocated enough training and staff to this stage in the first GL. The disadvantage was that we had a long period in which patient information was in different EHR systems. The protocols varied a lot in hospitals, which made, for example, patient transitions challenging.

Among other EPIC users in different countries, our professionals see APOTTI as a complex system that requires more documentation than the previous EHR systems [38]. One reason for the feeling of increased documentation may be the structured form of the information and the changes in the documenting process. For example, when the physicians now make an order straight to the medication list in structured form, it increases the documentation in the ordering phase, but saves work in the next steps of the process and improves medication safety, as the information is not transferred or transcribed manually by other professionals. To achieve closed-loop medication management, appropriate documentation for ordering and prescribing is crucial.

### 4.6. Limitations

In HUS, we saw that a few years was a very short period for implementing an EHR system based on closed-loop technology, and it required major changes in existing medication management and use processes. Our data describe the error reports six months after the last implementation; more data are needed on how ME trends change over a longer time period [34]. However, we needed these data urgently for the rapid continuing development of the APOTTI system. This was very important, because 43% of the reported severe MEs were linked to the APOTTI system.

As ME reports are based on voluntary, spontaneous, and anonymous reporting, the data have some limitations in describing medication safety in the organisation. There is a possibility that the trends in ME reports do not give us the whole picture of changes in medication safety. Before this study, we also evaluated other data available (e.g., IT support requests made by APOTTI users), but HaiPro data were the most usable and informative. We also changed the ME classification types during the study period, which made it challenging to evaluate some changes in the error trends. In HUS, we are now planning to also use data from APOTTI that will give us more information, for instance, about the real use of barcodes and MEs that have been detected.

Many interventions occurred simultaneously in this research, both in the new EHR system and technology and the changes in the medication processes. As also seen in our results, it was not always easy to evaluate whether the error was due to the new EHR system, a new user of the EHR system, or a mix of both. Likewise, it was not always clear if the reduction of some error types was because of the new EHR system or changes to the medication process that the new EHR system enabled. Some of the changes in our medication management process could have also been achieved in other EHR systems, with different solutions.

## 5. Conclusions

According to our study, implementation of a new EHR system can cause a temporary increase in medication error reports and significant changes in the error report profile. Implementing a structured EHR system requires many medication management process and workflow changes, which makes it complex, and it requires a strong focus on managing change. Our process of approaching a closed-loop EMMS with a new EHR system (APOTTI) seemed to provide safer medication dispensing and administration. However, medication reconciliation, ordering, and prescribing processes still need improvements in technical usability and user skill. This study highlights the importance of a proper usability evaluation before implementation, in order to allocate resources and successfully organise healthcare professionals’ training. Ordering and prescribing errors should be further studied to find strategies to minimise the MEs related to these. Furthermore, physicians would benefit from additional mandatory training on structured ordering and prescribing.

## Figures and Tables

**Figure 1 healthcare-10-01020-f001:**
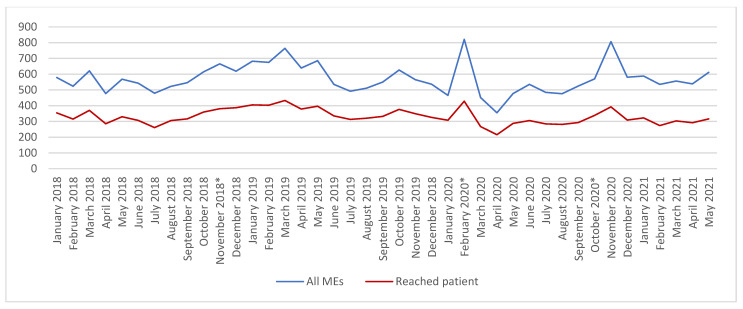
Trends in all reported MEs and MEs which reached patients during 1/2018–5/2021. * Different implantation phases (Go-Lives = GLs) GL1: 11/2018, GL2.1: 2/2020 and G 2.2. 10/2020.

**Figure 2 healthcare-10-01020-f002:**
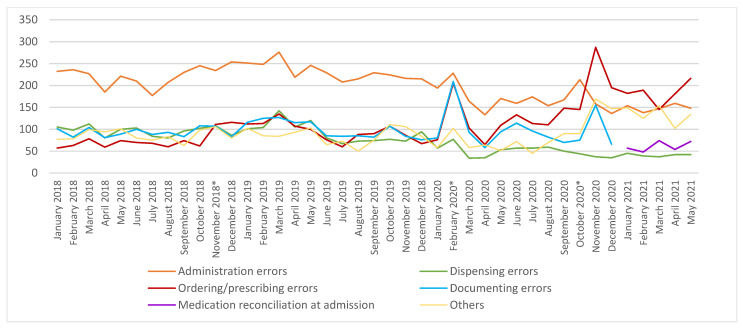
Trends in the medication error subtypes during 1/2018–5/2021. * Different implantation phases (Go-lives = GLs) GL1: 11/2018, GL2.1: 2/2020 and GL2.2. 10/2020. Others: Preparing and compounding, ordering drugs from the pharmacy, delivering drugs from the pharmacy, storage errors, unexpected reactions in a patient, not known, and monitoring errors (new type added in 2021). The medication reconciliation at admission error subtype was added, and documenting of the error subtype was removed from the HaiPro system at the beginning of 2021.

**Figure 3 healthcare-10-01020-f003:**
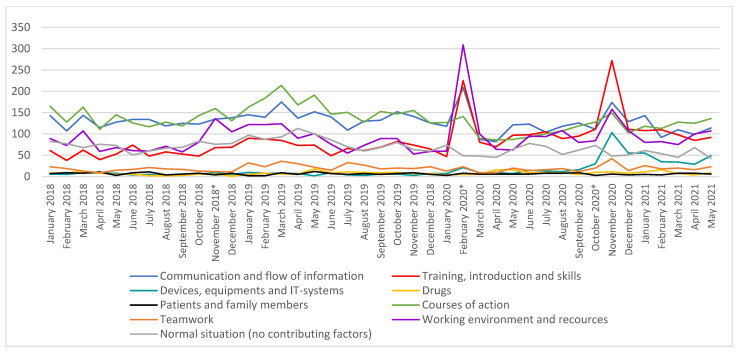
Trends in the reporter contributing factors related to MEs during 1/2018–5/2021. * Different implantation phases (Go-lives = GLs) GL1: 11/2018, GL 2.1: 2/2020 and GL 2.2. 10/2020. Contributing factors that were not included in the figure include organisation and management (only 0–5 reports per month) and not known contributing factors.

**Figure 4 healthcare-10-01020-f004:**
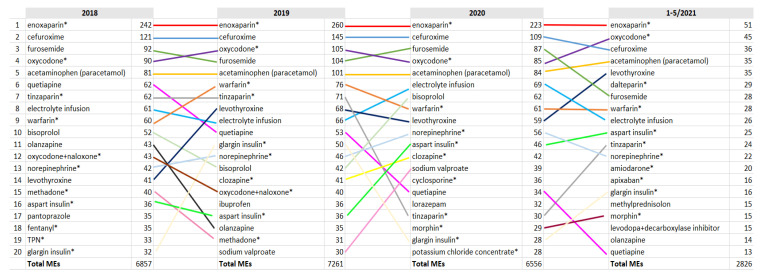
TOP20 medications related to the reported MEs in 2018–2020. * High-alert medication [7].

**Table 1 healthcare-10-01020-t001:** Key changes in the medication management process before and after implementing a new electronic health record (EHR) system (APOTTI).

Medication Process before APOTTI	Medication Process with APOTTI
Multiple EHRs in the hospital and medications are ordered in multiple systems.	There is one EHR and ordering system in the entire hospital.
Medication reconciliation in the EHR system is based only on hospital policy and documented in free text. Pharmacists are not widely involved in the process.	Medication reconciliation and a structured home-medication list are mandatory for in-patient medication. A home-medication list is integrated into the Kanta system, which holds electronic prescriptions [20]. Pharmacists are involved in medication reconciliation in many units.
Prescribing with free text orders and prescriptions in variable places in EHR systems.	Prescribing with structured order and prescription forms in specific medication applications in one EHR system.
Prescribing and ordering with the brand name.	Prescribing and ordering with the generic name.
Clinical decision support system (CDSS) for interactions and allergy warnings	More sophisticated CDSS, e.g., with dose warnings (including dosing with older patients and renal impairment), duplicate medications, and electronic best practice advice (BPA)
Primarily nurses transcribe orders to patients’ medication list. Verbal orders are common.	Primarily physicians document orders directly to patients’ medication list. Verbal orders are allowed only in limited situations.
Orders are not verified.	Pharmacists verify orders in some units.
Automated dispensing cabinets not integrated into the EHR system	Automated dispensing cabinets integrated with APOTTI enable the dispensing of medicines according to electronic orders.
Dispensing and preparing the medicines in units for the next shift or day (24 hours), some of the units use paper-based medication lists.	Dispensing and preparing the medicines in a timely manner (max. 2 hours before administration) by using the EHR system’s medication application and barcode scanning.
Medicines dispensed in the unit are double-checked by another nurse (manual double-check process).	Dispensing the right medicine is assured by scanning the barcodes of the medicine packages (no unit doses). A manual double-check process is used only when the barcode is not available or in use and for high alert medications (in addition to scanning).
Medicines prepared (e.g., dissolved and diluted) are double-checked in a few units, paper-based instructions for preparing medicines.	Preparing is documented by scanning the barcodes of the medicines, and EHR’s medication application gives the instructions for preparing. The manual double-check process is used only when the barcode is not available or in use and for the high alert medications (in addition to scanning).
The right patient and right medicine are assured manually when administering the medicine.	When administering the medicine, the right patient and medicines are assured by using the barcodes.
Medication administration is recorded with delay and only some of the medicines are recorded (e.g., high alert medications).	Medication administration is recorded in a timely manner at the bedside, and all medicines are recorded.

## Data Availability

Data are contained within the article.

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
