# Peer review of "Implementing a New Electronic Health Record System in a University Hospital: The Effect on Reported Medication Errors"

_healthcare, 2022, doi:10.3390/healthcare10061020_

Round 1

Reviewer 1 Report

There are some very mild English corrections needed.

The paper introduces an implementation of a closed loop electronic medication management system at the Helsinki University Hospital System. 

To evaluate the impact of (APOTTI system) a closed loop medication management system in the EPIC emr at their institution on reported medication errors.

The conclusions show temporary changes associated with the new implementation followed by a shift in the data reported by physicians, and a change in the medication error types captured some due to the process change itself and some due to increased visibility of specific issues.

The data also identified concerns with medication reconciliation and alert fatigue that need to be overcome to improve the outcomes of this system.

The study is limited to one health system and occurred during the pandemic which may have impacted the consistent staffing.

The time frame is short which is acknowledged in the article.

No major errors, just slight differences in sentence structure for example :

Especially prescribing, administering, monitoring and transition of care are the most error-prone phases for severe MEs in the medication process.  (Especially, not needed or Specifically better choice)

These do not impede comprehension of the article.

Author Response

Comment: There are some very mild English corrections needed.

The paper introduces an implementation of a closed loop electronic medication management system at the Helsinki University Hospital System. 

To evaluate the impact of (APOTTI system) a closed loop medication management system in the EPIC emr at their institution on reported medication errors.

The conclusions show temporary changes associated with the new implementation followed by a shift in the data reported by physicians, and a change in the medication error types captured some due to the process change itself and some due to increased visibility of specific issues.

The data also identified concerns with medication reconciliation and alert fatigue that need to be overcome to improve the outcomes of this system.

The study is limited to one health system and occurred during the pandemic which may have impacted the consistent staffing.

The time frame is short which is acknowledged in the article.

No major errors, just slight differences in sentence structure for example:

Especially prescribing, administering, monitoring and transition of care are the most error-prone phases for severe MEs in the medication process.  (Especially, not needed or Specifically better choice)

These do not impede comprehension of the article.

Authors’ response: Thank you for your valuable and positive comments. We are sorry to hear that there are some English corrections needed. The grammar of the manuscript has been professionally edited. We have made changes in language according to the specific comments given by the Reviewer. The word especially (line 35) has been removed.

Reviewer 2 Report

The manuscript describes the impact of implementation of a new electronic medication system in a health care system on medication errors.  The findings and discussions are relevant and will be beneficial to other health care systems who are exploring or beginning to implement electronic medication management systems.  The manuscript is very well-written and is a nice summary of abundant data.  

1.On lines 78 and 106, acronyms are used that have not been written out previously (ATC and ISMP, respectively).  With those minor corrections, I recommend the manuscript for publication.

2.Some detail on the contributing factors in Figure 3 would be helpful. Many are self-explanatory ("Patients and family members", "Drugs") but others are less clear ("Teamwork" or "Courses of action").

3.An additional limitation to be considered and addressed would be HaiPro being voluntarily reported. What, if any, impact was that on the findings?

4.Some description of previous ME rates from a comparable period of time pre-implementation would be beneficial.

Author Response

General comment: The manuscript describes the impact of implementation of a new electronic medication system in a health care system on medication errors.  The findings and discussions are relevant and will be beneficial to other health care systems who are exploring or beginning to implement electronic medication management systems.  The manuscript is very well-written and is a nice summary of abundant data. 

Authors’ response: Thank you for these positive comments on our manuscript. 

Comment 1: On lines 78 and 106, acronyms are used that have not been written out previously (ATC and ISMP, respectively).  With those minor corrections, I recommend the manuscript for publication.

Authors’ response: Thank you for pointing out these errors, these acronyms have now been written out.

Comment 2: Some detail on the contributing factors in Figure 3 would be helpful. Many are self-explanatory ("Patients and family members", "Drugs") but others are less clear ("Teamwork" or "Courses of action").

Authors’ response: Thank you for this comment. These are based on structured classification in the HaiPro tool. In HaiPro tool there are no definitions for the categories of contributing factors, but they include subcategories. We are not able to provide this on manuscript text or figure 3. as all contributing factors include subcategories and they are multiple. If needed, we can provide a supplementary file for the manuscript where these categories and their subcategories would be presented in more detailed.

Comment 3: An additional limitation to be considered and addressed would be HaiPro being voluntarily reported. What, if any, impact was that on the findings?

Authors’ response: Thank you for this valuable comment. We have added word voluntary on Limitations section (line 341).

Comment 4: Some description of previous ME rates from a comparable period of time pre-implementation would be beneficial.

Authors’ response: Thank you for this comment. For this purpose, we have presented the error data almost a year (from January 2018) before first implementation in November 2018 in the Figures 1-3. The ME rates before year 2018 are comparable to year 2018 (no as clear changes in ME categories) and we think longer period of time wouldn´t add any relevant information to the study.

Reviewer 3 Report

This article has an interesting idea, but it does not have a good writing structure and needs a basic review as follows:

-The abstract need to be revised and improved to reflect the main idea.

- Keywords need to be improved.

- In the introduction of the article, mention other healthcare systems. For example, cite the following articles in the introduction:

1- Sengan, S., Khalaf, O.I., Sharma, D.K. and Hamad, A.A., 2022. Secured and privacy-based IDS for healthcare systems on E-medical data using machine learning approach. International Journal of Reliable and Quality E-Healthcare (IJRQEH), 11(3), pp.1-11.

2- Elhadad A, Alanazi F, Taloba AI, Abozeid A. Fog Computing Service in the Healthcare Monitoring System for Managing the Real-Time Notification. Journal of Healthcare Engineering. 2022 Mar 15;2022.

3- Syed, S.A., Sheela Sobana Rani, K., Mohammad, G.B., Chennam, K.K., Jaikumar, R., Natarajan, Y., Srihari, K., Barakkath Nisha, U. and Sundramurthy, V.P., 2022. Design of Resources Allocation in 6G Cybertwin Technology Using the Fuzzy Neuro Model in Healthcare Systems. Journal of Healthcare Engineering, 2022.

- Please pay attention to the grammar of the English language and try to improve it in general.

- References are not formatted correctly.

- In conclusion must explain the advantages and disadvantages of this framework based on the results to become this manuscript as guide of other researchers. therefore, conclusion must be revised.

Author Response

Comment 1: This article has an interesting idea, but it does not have a good writing structure and needs a basic review as follows:

-The abstract need to be revised and improved to reflect the main idea.

- Keywords need to be improved.

Authors’ response: Thank you for this comment but we are not quite sure what should be improved in our abstract and keywords. We see that the abstract and key words are in line with the contents of the manuscript. We kindly ask you to provide specific examples which part of the abstract or keywords need improvements. We would be happy to make these corrections.

- In the introduction of the article, mention other healthcare systems. For example, cite the following articles in the introduction:

1- Sengan, S., Khalaf, O.I., Sharma, D.K. and Hamad, A.A., 2022. Secured and privacy-based IDS for healthcare systems on E-medical data using machine learning approach. International Journal of Reliable and Quality E-Healthcare (IJRQEH), 11(3), pp.1-11.

2- Elhadad A, Alanazi F, Taloba AI, Abozeid A. Fog Computing Service in the Healthcare Monitoring System for Managing the Real-Time Notification. Journal of Healthcare Engineering. 2022 Mar 15;2022.

3- Syed, S.A., Sheela Sobana Rani, K., Mohammad, G.B., Chennam, K.K., Jaikumar, R., Natarajan, Y., Srihari, K., Barakkath Nisha, U. and Sundramurthy, V.P., 2022. Design of Resources Allocation in 6G Cybertwin Technology Using the Fuzzy Neuro Model in Healthcare Systems. Journal of Healthcare Engineering, 2022.

Authors’ response: We want to thank you for introducing these interesting references which were new for us. We have read them but unfortunately these do not address electronic medication management systems, medication errors or medication safety, which is in focus of this study. We would appreciate Reviewer´s suggestions how these references could be linked to our study to support the contents of our introduction section. There are many electronic health record systems (also in Finland) and we focus on this article on EPIC based APOTTI without even trying to compare it to other systems.

Comment 2: Please pay attention to the grammar of the English language and try to improve it in general.

Authors’ response: We are sorry to hear this because the grammar of the manuscript has been professionally edited. We kindly ask you to provide specific examples which need improvements. We would be happy to make these corrections.

Comment 3: References are not formatted correctly.

Authors’ response: Thank you for noticing this. We have carefully checked the reference list and made some corrections. We kindly ask you to provide specific examples which still need improvements and we would be happy to make these corrections.

Comment 4: In conclusion must explain the advantages and disadvantages of this framework based on the results to become this manuscript as guide of other researchers. therefore, conclusion must be revised.

Authors’ response: Thank you for this comment. We have explained these at the discussion section, especially at the limitations section 4.6. If this is still not seen sufficient, we are happy to reconsider this comment.

Round 2

Reviewer 1 Report

appreciate edits for grammar thank you

Reviewer 2 Report

All of my concerns have been addressed.

Reviewer 3 Report

The manuscript can be accepted. No further comments.